# Attention Based Variational Graph Auto-encoder (AVGAE)

## Abstract

Recently techniques such as VGAEs (Variational Graph Autoencoder) are quite popular in the unsupervised task setting and in generative modeling. Unlike conventional autoencoders, which typically use fully-connected layers to learn a latent representation of input data, VGAEs operate on graph-structured data. We propose to incorporate attention in VGAEs (AVGAE) for capturing the relationships better thereby increasing the robustness and generalisability. In a VAE, the encoder network learns to map input data to a lower-dimensional latent space, while the decoder network learns to map latent space vectors back to the original input data. Unlike traditional autoencoders, which typically use a fixed encoding function, VAEs use a probabilistic encoding function that maps input data to a probability distribution over the latent space. They have been shown to improve the quality of the generated output, particularly for tasks where the input data is complex and high-dimensional.

## 1 Attention Based VGAE

We build up on the variational graph autoencoder (VGAE) by making use of multi-head self-attention to parameterize the mean and standard deviation of the latent variable distribution. In an attention-based VGAE, the encoder network uses an attention mechanism to selectively focus on parts of the input data that are most relevant for the current encoding step and a simple inner product decoder. This allows the model to learn more informative representations of the data, which can aid in generating a better output.

### 1.1 Notations

Denote an undirected graph $\mathcal{G} = (\mathcal{V}, \mathcal{E})$ with $N$ nodes. Denote the adjacency matrix of $\mathcal{G}$ as $\mathbf{A}$ and its degree matrix as $\mathbf{D}$. Denote $z_i$ as the stochastic latent variables for each node $i$ with node feature $x_i$. The latent variables and node features are denoted by matrices $\mathbf{Z} \in \mathbb{R}^{N \times F}$ and $\mathbf{X} \in \mathbb{R}^{N \times D}$.

### 1.2 Encoder Model with attention

We propose the encoder model as

$$q(\mathbf{Z}|\mathbf{X}, \mathbf{A}) = \prod_{i=1}^{N} q(z_i|\mathbf{X}, \mathbf{A}) \tag{1}$$

and

$$q(z_i|\mathbf{X}, \mathbf{A}) = \mathcal{N}(z_i|\mu_i, \mathrm{diag}(\sigma_i^2)) \tag{2}$$

Here

$$\mu_i = \Big\|_{m=1}^{M} \Big[ \sum_{v_k \in \mathcal{V} \setminus \{x_i} \alpha_{i,k}^{(m)} \, \mathrm{GCN}_\mu(x_k)} \Big] \tag{3}$$

$$\log \sigma_i = \Big\|_{m=1}^{M} \Big[ \sum_{v_k \in \mathcal{V} \setminus \{x_i\}} \beta_{i,k}^{(m)} \, \mathrm{GCN}_\sigma(x_k) \Big] \tag{4}$$

where $M$ denotes the number of attention heads, $\text{GCN}_\mu$ and $\text{GCN}_\sigma$ are denotes a two-layer GCNs defined as $\text{GCN} = \tilde{\mathbf{A}}\,\text{ReLU}(\tilde{\mathbf{A}}\mathbf{X}W_0)W_1$ with weights $W_i$ and $\tilde{\mathbf{A}}$ denoting the symmetrical normalized adjacency matrix of $\mathcal{G}$. More details about the decoder can be found in appendix A.1

### 1.3 TRAINING OBJECTIVE

Similar to the case of VGAE, we optimize the variational lower bound $\Gamma$ with respect to the parameters $W_i$, given by

$$\Gamma \;=\; \mathbb{E}_{q(\mathbf{Z}|\mathbf{X},\mathbf{A})}[\log p(\mathbf{A}|\mathbf{Z})] - \text{KL}[q(\mathbf{Z}|\mathbf{X},\mathbf{A})\|p(\mathbf{Z})] \tag{5}$$

Here $\text{KL}[f(\cdot)\|g(\cdot)]$ denotes he Kullback-Leibler divergence between $f(\cdot)$ and $g(\cdot)$. We assume a Gaussian prior for $p(\mathbf{Z}) = \prod_i p(z_i) = \prod_i \mathcal{N}(z_i\,|\,0,\mathbf{I})$

## 2 EXPERIMENTS AND RESULTS

We train our model on Cora dataset and the Citeseer dataset (Sen et al., 2008) for link prediction and compare it with classic VGAE. We trained our model using a cyclic learning rate scheduler with learning rates in $(0.0003, 1)$ and SGD optimizer for 3000 epochs. We use a hidden space dimension of 32 and a latent space dimension of 16. We use 2 attention heads.

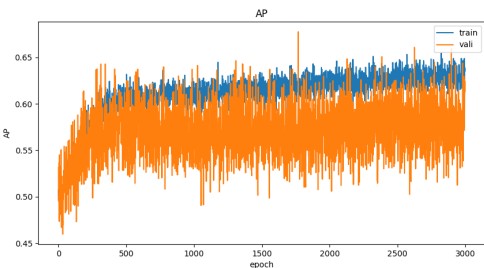

Figure 1: Average Precision vs epochs for Cora Dataset

Table 1: Link prediction task in citation networks. See Sen et al. (2008) for dataset details.

| Method | Cora | | Citeseer | |
|---|---|---|---|---|
| | AUC | AP | AUC | AP |
| GAE | $61.0 \pm 0.02$ | $62.0 \pm 0.03$ | $59.5 \pm 0.04$ | $59.9 \pm 0.05$ |
| VGAE | $61.4 \pm 0.01$ | $62.6 \pm 0.01$ | $60.8 \pm 0.02$ | $62.0 \pm 0.02$ |
| AVGAE | $\mathbf{69.5 \pm 0.03}$ | $\mathbf{70.2 \pm 0.03}$ | $\mathbf{65.3 \pm 0.02}$ | $\mathbf{65.7 \pm 0.01}$ |

We use average precision (AP) and area under the ROC curve (AUC) as the metric, as indicated in figure 1. The models are trained on an incomplete version of these datasets, some parts of the citation links that is the edges are removed, while the corresponding node features are retained.

## 3 CONCLUSION

This paper depicts the importance of using attention in a graph-like architecture, an attention mechanism that computes a relevance score for each node in the graph based on its features and the features of its neighbors. Variational autoencoder tries to learn the probabilistic distribution of the input data. Using attention in the encoder will help to learn the dependency of nodes amongst themselves instead of treating them as independent and identically distributed (iid) random variables. It can also be inferred from the results in table 1 that attention strengthens the representation power of the encoder.

URM STATEMENT

The authors acknowledge that at least one key author of this work meets the URM criteria of ICLR 2023 Tiny Papers Track.

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

## A  APPENDIX

### A.1  DECODER MODEL A.K.A GENERATIVE MODEL

We use the same decoder model as in Kipf & Welling (2016).

$$p(\mathbf{A}|\mathbf{Z}) \quad = \quad \prod_{i=1}^{n} \prod_{j=1}^{N} p(A_{ij}|z_i, z_j) \text{ with } p(A_{ij} = 1|z_i, z_j) = \text{sigmoid}(z_i^\top z_j) \qquad (6)$$

where $A_{i,j}$ denotes the entry in $i^{\text{th}}$ row and $j^{\text{th}}$ column of $\mathbf{A}$.

### A.2  ADDITIONAL FIGURES

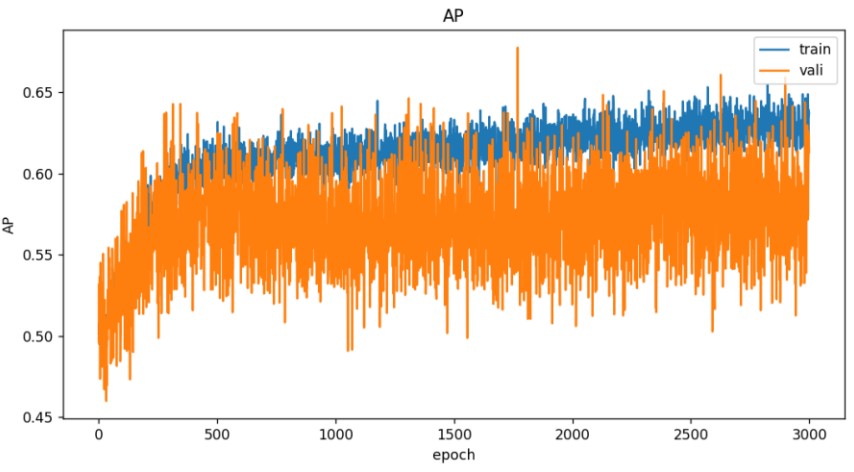

Figure 2: Average Precision vs epochs for Citeseer dataset

