# OpenReview forum: "Attention Based Variational Graph Auto-Encoder (AVGAE)"
_ICLR.cc/2023/TinyPapers — Submitted to Tiny Papers @ ICLR 2023_

### Official Review · Reviewer_H7n8 · 2023-03-23

**Confidence:** 4

**Summary Of Contributions:**

The paper proposes Attention Based Variational Graph Auto-Encoder (AVGAE). AVGAW involves two improvements compared to the conventional Graph Auto-Encoder  which are using probabilistic encoder (from VAE) and using attention architecture as replacement for the conventional fully connected architecture.

**Rating:**

Clear, Correct, and Reproducible (CCR): a submission which meets the reviewing criteria

**Strengths And Weaknesses:**

# Strengths
- The paper is new in terms of applying probabilistic encoder and attention architecture to GAE.
- Experiments show the benefit of the proposed AVGAE compared to the GAW and the VGAE.

# Weaknesses
- The mathematical derivation should be written in more detail instead of high-level equations e.g., parameterization trick, Jensen inequality

**Suggested Changes:**

The mathematical derivation should be written in more detail instead of high-level equations e.g., parameterization trick, Jensen inequality.

---

### Official Review · Reviewer_USk2 · 2023-03-28

**Confidence:** 5

**Summary Of Contributions:**

This work introduces GAT-style attention into a Variational Graph Autoencoder (method is called AVGAE) as opposed to the standard convolutional approach. Method optimises standard ELBO while outperforming GAE and vanilla VGAE on Cora and Citeseer.

**Rating:**

Great Start (GS): a submission which meets some of the reviewing criteria but has room for improvement

**Strengths And Weaknesses:**

Hello authors,

Thank you for choosing to participate in Tiny Papers this year! Here are some comments:

**STRENGTHS**
- Clear-cut explanation of the method and what it brings to the table. While not novel, the work provides a systematic recount of the different moving parts involved.
- Evident that the method introduced (AVGAE) outperforms its no-attention counterparts.

**WEAKNESSES**
- Cora and Citeseer, despite being in the community for so long, are not great benchmark datasets as they lack the problem complexity needed to truly evaluate GNN performance with [[https://arxiv.org/abs/2003.00982](https://arxiv.org/abs/2003.00982)]. Using them as the target datasets doesn't guarantee true, objective improvement given their simplicity.

**Suggested Changes:**

Hello authors, good work on the paper. Here are some pressing changes and good-to-haves to solidify the paper:

**URGENT**
- Equation (3) seems to be unformatted. I was able to make out what it means because of Equation (4).
- There was no explanation given as to why you used incomplete versions of the datasets (page 2, paragraph below table) or why certain edges in the citation graphs were removed. A short line or two justifying this design choice would be good.
- Typo: "the" instead of "he" in the Training Objective second paragraph ("Here KL[.|.] denotes **the** Kullback-Leibler ...").
- I would suggest benchmarking on high-quality datasets such as ZINC, MNIST, PATTERN, and CLUSTER. You can find them here: [https://github.com/graphdeeplearning/benchmarking-gnns](https://github.com/graphdeeplearning/benchmarking-gnns).

**GOOD-TO-HAVE**
- Ablate how # of heads affects performance (more attention = better AP?)

---

### Comment · Area_Chair_8XVQ · 2023-06-06
**Archival**

 This work does not meet the threshold for archival in terms of CCR.

---

### Meta-Review · Area_Chair_8XVQ · 2023-04-06

**Recommendation:** Invite to archive
**Confidence:** 4

**Metareview:**

- Clarity: The method explanation is somehow clear, but the equations look unformatted, and the paper does not provide a proper literature review, which might create difficulties for the reader to place the work among others correctly
- Correctness: Experiments seem valid; even if more relevant datasets can be considered, for the space and the purpose of the paper, the experiments already highlight interesting performance gain
- Reproducibility: Given the presentation issues and the lack of details, it might be complicated to reproduce the experimental setting. The paper does not mention code release.

Overall, the reviewers express satisfaction with the proposed approach but are concerned about the mathematical formulation that should be revisited.

**Summary:**

The paper proposes an attention-based architecture to learn on graphs. The approach is considered relevant, but the presentation raises doubts, especially regarding method reproducibility.

**Comments And Feedback To The Authors:**

I appreciate the work of the authors, and overall it is indeed a great start, which might also reach a good impact if the presentation is clarified and the experiments a bit more developed. Providing the correct literature to contextualize the paper is particularly important: it helps the reader to understand what is the focus, sharpening the message of the paper and its contribution. I suggest considering the reviewer's suggestions and working around the math and notation more. An option is to expand the appendix, providing the needed mathematical tools and derivations to make the paper more self-contained. Also, consider providing more details about the network architecture and the hyperparameters for reproducibility and possibly release the code.

**Reason For Not Giving A Higher Recommendation:**

The paper presentation requires some significant effort, especially in the mathematical formulation. The paper cites only two previous works and does not discuss alternatives or work placement in the field.

**Reason For Not Giving A Lower Recommendation:**

The method is considered overall enjoyable, pointing in a good direction. Experiments, even if preliminary, prove some benefit and might foster the discussion. The topic seems orthogonal to different applicative fields and can collect attention from ICLR community.

---

### Decision · Program_Chairs · 2023-04-08

Invite to archive